# Advances in Immunotherapy and Targeted Therapy of Malignant Melanoma

**DOI:** 10.3390/biomedicines13010225

**Published:** 2025-01-17

**Authors:** Xue Wang, Shanshan Ma, Shuting Zhu, Liucun Zhu, Wenna Guo

**Affiliations:** 1School of Life Sciences, Zhengzhou University, Zhengzhou 450001, China; wangxue2588@163.com (X.W.); mashanshan84@163.com (S.M.); zstlyt@163.com (S.Z.); 2School of Life Sciences, Shanghai University, Shanghai 200444, China; zhuliucun@shu.edu.cn

**Keywords:** malignant melanoma, immunotherapy, targeted therapy, inhibitors, combination therapy, precision medicine

## Abstract

Malignant melanoma (MM) is a malignant tumor, resulting from mutations in melanocytes of the skin and mucous membranes. Its mortality rate accounts for 90% of all dermatologic tumor mortality. Traditional treatments such as surgery, chemotherapy, and radiotherapy are unable to achieve the expected results due to MM’s low sensitivity, high drug resistance, and toxic side effects. As treatment advances, immunotherapy and targeted therapy have made significant breakthroughs in the treatment of MM and have demonstrated promising application prospects. However, the heterogeneity of tumor immune response causes more than half of patients to not benefit from clinical immunotherapy and targeted therapy, which delays the patient’s condition and causes them to suffer adverse immune events’ side effects. The combination of immunotherapy and targeted therapy can help improve therapeutic effects, delay drug resistance, and mitigate adverse effects. This review provides a comprehensive overview of the current development status and research progress of immune checkpoints, targeted genes, and their inhibitors, with a view to providing a reference for the clinical treatment of MM.

## 1. Introduction

Malignant melanoma (MM) accounts for 90% of mortality from all cutaneous tumors [1]. MM is one of the most aggressive metastatic cancers and can spread from a relatively small primary tumor and metastasize to multiple sites, including the lungs [2], liver, brain [3], lymph node [4], and bones [5], and the 5-year survival rate for metastatic MM is only 10% [6]. The spread of MM cells to these organs often leads to the development of multiple organ failure and ultimately contributes to the high mortality rate [7]. An estimated 331,647 new cases and 58,645 deaths of MM occurred in 2022 [8]. Traditional treatments for MM, including radiation, chemotherapy, and surgery, are poorly targeted and result in poor patient prognosis [9,10]. Early primary MM can be treated with surgical resection, but metastatic MM has metastasized to other organs and cannot be surgically removed [7]. In contrast, recent advances in immunotherapy and targeted therapy offer new hope for enhancing treatment efficacy [11]. Immunotherapy and targeted therapy have emerged as important treatment modalities in recent years [12]. Immunotherapy, which aims to enhance the body’s immune response against cancer cells, and targeted therapy, which focuses on specific molecular targets within cancer cells, have shown certain positive effects [13,14]. Immunotherapy has been a major driving force in MM treatment for decades, despite historically poor clinical outcomes [9,10]. Targeted therapies, which attack specific gene mutations, pathways, or proteins associated with the development of MM, have also been pivotal [15]. Metastatic melanoma survival has also gotten better in recent years because of progress in targeted therapies and immunotherapies [16,17]. Still, long term survival is restricted. Recent data show that the 5-year survival rate for metastatic melanoma is about 22.5%, but this number changes based on the metastases’ location and the tumor burden [18,19]. Over the past few decades, a range of immunotherapy and targeted therapy drugs, along with combination approaches, have been approved by the US Food and Drug Administration (FDA) (Figure 1) [15,20]. Although these current therapies have improved the prognosis of patients with MM, challenges such as low efficacy rates and inevitable treatment resistance remain. This review paper summarizes the relevant literature to provide a reference for achieving precise treatment of MM.

## 2. Immunotherapy and Inhibitors

Immune checkpoints are cell-surface proteins expressed by immune cells, the function of which is to control the initiation, duration, and magnitude of immune responses, and particularly relevant to T-cell function (Figure 2) [12]. Immune checkpoint inhibitors (ICIs) are a type of immunotherapy that work by blocking proteins on immune cells or tumor cells that prevent the immune system from attacking cancer cells. These proteins are normally involved in regulating the immune response to prevent it from attacking healthy cells [21]. However, tumor cells can use these checkpoints to evade the immune system. MM cells exhibit substantial immunogenicity. The discovery of immune checkpoint proteins such as PD-1/PDL-1 and CTLA-4 was a major breakthrough in the field of cancer immunotherapy. Humanized monoclonal antibodies against these immune checkpoint proteins have been successfully used in patients with MM. ICIs improve therapeutic response by increasing the sensitivity of the immune system to tumor cells [22]. The efficacy of ICIs varies across melanoma subtypes [23,24]. Currently, ICIs, particularly those targeting PD-1, CTLA-4, and LAG-3, have demonstrated superior efficacy compared to conventional therapies in treating MM (Figure 3) [25,26,27].

### 2.1. Programmed Cell Death Protein-1 (PD-1) Inhibitors

PD-1 is a member of the immunoglobulin superfamily and is primarily expressed on immune cells such as macrophages, dendritic cells, natural killer cells, T cells, and B cells [28]. PD-1 plays a critical role in programmed death signaling and regulates T cell-mediated responses. When PD-1 bind to its ligands, it disrupts the downstream signaling pathways essential for T-cell activation and inhibits transcription, thereby suppressing T-cell immune responses [29,30]. Currently, the primary anti-PD-1 monoclonal antibodies available are nivolumab and pembrolizumab. These antibodies specifically target the interactions between PD-1 and its ligands PD-L1 and PD-L2, setting the stage for a detailed discussion of their roles in the treatment of malignant melanoma [31,32].

Nivolumab, a humanized immunoglobulin monoclonal antibody, has been approved by the FDA as a single agent for patients with BRAF V600 wild-type unresectable or metastatic MM [33]. Clinical trials have demonstrated that in patients with stage IIIB/C or stage IV melanoma, those who received adjuvant treatment with nivolumab and the anti-CTLA-4 antibody Ipilimumab, after surgical resection of tumors, experienced significantly higher 12-month progression-free survival (RFS) compared to those treated only with ipilimumab. Additionally, the incidence of grade 3–4 side effects was significantly lower in the Nivolumab-treated group [34]. Because Nivolumab has shown significant effects in the treatment of patients with MM, the FDA has approved it for use in the adjuvant treatment of patients with advanced lymph node dissection or metastatic MM.

Pembrolizumab, an IgG4-kappa monoclonal antibody, is the first anti-programmed-death-1 (PD-1) drug licensed by the FDA for the treatment of advanced MM [35]. Pembrolizumab has demonstrated durable antitumor activity and tolerability in the treatment of patients with advanced MM [36]. Pembrolizumab also prolongs PFS and overall survival (OS) in patients with advanced MM, with lower toxicity than Ipilimumab [37]. In addition, clinical trial results showed that Pembrolizumab significantly reduced the risk of disease recurrence or death as an adjuvant treatment for stage IIB or IIC MM [38].

### 2.2. Cytotoxic T-Lym-Phocyte Antigen 4 (CTLA-4) Inhibitors

CTLA-4, a member of the immunoglobulin-associated receptor family, is expressed solely on T cells and governs the amplitude of T-cell activation throughout the early phases. CTLA-4 primarily inhibits the function of CD28 [39,40]. CTLA-4 inhibits T-cell activation and enhances the immunosuppressive activity of T regulatory cells, acting as a negative regulator in the immune process [41]. Currently, anti-CTLA-4 antibodies mainly include Ipilimumab and Tremelimumab.

Ipilimumab is the first FDA-approved specific antibody that can inhibit the function of CTLA-4, and its emergence provides new possibilities for immune checkpoint blockade (ICB) therapy [42,43]. Compared to interferon, treatment with Ipilimumab significantly prolongs OS in postoperative stage III MM patients with a high risk of cutaneous recurrence [44]. Ipilimumab promotes T-cell activation and proliferation, and it can kill MM cells by activating the immune system. However, it also can cause other immune-related side effects. It has been noted that the proportion of adverse events resulting in treatment discontinuation after treatment with high doses of Ipilimumab is about 53%, and most of them are grade 3–4 adverse events [45]. Since CTLA-4 is involved in autoimmune prevention as a negative regulator of T-cell immune responses, blockade of it by ibritumomab may lead to immune-related adverse effects (irAEs), such as colitis and enterocolitis [46]. Further studies are needed to investigate the therapeutic limitations and resistance mechanisms of Ipilimumab-mediated immunotherapy.

Tremelimumab was the first anti-CTLA-4 antibody to be investigated. In the results of a clinical trial, it was noted that treatment with Tremelimumab significantly improved the duration of response to the drug in patients with inoperable stage III/IV MM compared to the use of standard chemotherapeutic agents (Temozolomide or Dacarbazine). Due to the similar objective response rate of patients in the Tremelimumab-treatment group and the final death of 340 patients, the trial was stopped, and the effectiveness of Tremelimumab in the treatment of MM was not successfully confirmed. However, the duration of response was longer than that in the chemotherapy group, which also had certain clinical significance [47].

### 2.3. Lymphocyte Activation Gene-3 (LAG-3) Inhibitors

LAG-3 is a type I transmembrane protein, highly structurally homologous to CD4, and is mainly expressed on CD4+ T cells, CD8+ T cells, natural killer cells, and Treg cells [48]. Previous studies have shown that blocking LAG3 increased the proliferation of CD4+ and CD8+ T cells [46]. Anti-PD-1 antibodies could only activate T cells but not inhibit regulatory T-cell activity, whereas anti-LAG-3 antibodies not only restored T-cell function but also inhibited regulatory T-cell activity. Therefore, in clinical development, anti-LAG-3 antibodies are often used in combination with anti-PD-1/PD-L1 antibodies. The combination of the LAG-3 monoclonal antibody Relatlimab and the anti-PD-1 antibody Nivolumab is effective in prolonging PFS and progression-free survival rate in patients with unresectable or metastatic MM [49]. As a result, Relatlimab, a LAG3-blocking antibody, combined with Nivolumab for the treatment of unresectable or metastatic MM, was approved by the FDA in 2022 [50].

Although only one of the anti-LAG-3 antibodies has been approved, several anti-LAG-3 antibodies have entered clinical studies, such as [51], BI 754111 [52], and Ieramilimab [53], and the combination of IMP321 with Pembrolizumab (NCT02676869). Information on safety and efficacy of these antibodies is limited.

Despite immunotherapy having achieved remarkable success in MM treatment, it also has certain drawbacks. Firstly, ICIs may lead the immune system to attack other organs in the body, resulting in irAEs like pneumonia, hepatitis, and endocrine diseases [54,55]. The immune activation that underlies most irAEs might be coupled with the activity required for antitumor immune responses (Figure 4) [56]. Secondly, there is the issue of drug resistance. Some patients may become resistant to ICIs, which causes a decline in treatment effectiveness [57,58]. Furthermore, the high cost of these inhibitors may limit their use among some patients. Future studies need to further explore ways to enhance the efficacy of these drugs, reduce adverse effects, and cut down the treatment cost.

## 3. Targeted Therapy and Inhibitors

Targeted therapies are drugs that target specific genetic mutations, pathways, or proteins associated with cancer development. Compared to conventional chemotherapy, the inhibitors used in targeted therapy can pinpoint mutated tumor genes, and these inhibitors do not harm the normal tissue cells surrounding the tumor, making targeted therapy more accurate and effective compared with other treatment [59]. Currently, targeted therapy for MM mainly uses two types of monoclonal antibodies, BRAF and MEK. BRAF inhibitors can inhibit activated BRAF-mutated kinase, and MEK inhibitors can inhibit the activation of MEK (Figure 5).

### 3.1. V-Raf Murine Sarcoma Viral Oncogene Homolog B1 (BRAF) Inhibitors

Activating mutations in BRAF occur in about 50% of skin melanomas [45]. The BRAF protein consists of three conserved structural domains. When the Valine at position 600 of the BRAF protein is replaced by a Glutamate, a BRAF V600E mutation occurs, leading to a significant increase in BRAF kinase activity, which activates the MAPK signaling pathway and thus transforms melanocytes into MM cells [60,61]. BRAF inhibitors are the first targeted therapeutic agents for patients with MM [48], and currently the most representative BRAF inhibitors are Vemurafenib [62], Dabrafenib [63], and Encorafenib [64].

Vemurafenib reduces the tumor size in more than half of advanced BRAF V600E mutation MM patients, while also keeping patients stable [63]. The results confirmed that the Vemurafenib-treatment group significantly prolonged patients’ PFS [65]. Moreover, compared to the chemotherapy group, BRAF V600E mutant metastatic MM patients with long-term survival in the Vemurafenib-treatment group did not frequently experience adverse skin events [66]. Although Vemurafenib has shown significant effects in the pre-treatment of patients with BRAF V600E mutation, patients are prone to develop resistance to it, such that 80% of MM patients treated for 3 years have shown resistance to Vemurafenib [67]. To better address this issue, recent findings suggest that the sensitivity of BRAF V600E mutant MM patients to Vemurafenib-treatment can be improved by targeting Glutathione Peroxidase 4 (GPX4) and ferroptosis through the plant sesquiterpene lactones DET and DETD-3 [68].

Compared with the Dacarbazine-chemotherapy group, the Dabrafenib-treatment group not only improved the PFS of patients with BRAF V600E mutant metastatic MM but also increased the partial emission rate (PR) and complete remission (CR) rates. Therefore, in 2013, the FDA approved Dabrafenib as a single agent in the treatment of MM patients with BRAF V600E mutation [69]. In addition, a trial validated that BRAF inhibitors combined with MEK inhibitors could improve the OS of metastatic MM patients with BRAF mutations [70]; so, a year later, the FDA approved Dabrafenib in combination with the MEK inhibitor Trametinib for MM treatment. A recent clinical trial again validated that Dabrafenib combined with Trametinib treatment significantly improved PFS and PR in advanced MM patients [71].

Encorafenib has also been approved by the FDA for the treatment of MM because of its ability to bind to mutant BRAF receptors and its long dissociation half-life [72]. The results of a clinical trial conducted in 2018 showed that PFS was increased approximately twofold in MM patients treated with the combination of Encorafenib and the MEK inhibitor Binimetinib [73]. The latest clinical trial demonstrated yet another improvement in OS in patients with BRAF V600E mutant MM treated with Encorafenib in combination with Binimetinib [74].

### 3.2. Mitogen-Activated Protein (MEK) Inhibitors

MEK is a mitogen-activated protease downstream of BRAF [75], and MEK inhibitors reduce the activity of the MAPK signaling pathway by blocking the expression of MEK genes located downstream of BRAF genes and RAS genes, thereby inhibiting tumor cell proliferation [76]. Trametinib, Binimetinib, and Cobimetinib are the main MEK inhibitors approved by the FDA for combination therapy or single therapy of metastatic MM.

Trametinib is orally effective, has a long half-life, and is well tolerated. The results of a phase III clinical trial showed that Trametinib improved PFS and OS in advanced MM patients compared to treatment with Dacarbazine or Paclitaxel alone [77]. However, MEK inhibitors have greater side effects when treating MM alone. In order to reduce the side effects and improve the therapeutic efficacy of MEK inhibitors during treatment, combining MEK inhibitors remains the focus of current research. For example, the BRAF inhibitor Vemurafenib combined with the MEK inhibitor Cobimetinib significantly improved PFS and PR in locally advanced or metastatic MM patients with BRAF V600E mutation [78].

### 3.3. V-Kit Hardy-Zuckerman 4 Feline Sarcoma Viral Oncogene Homolog (KIT) Inhibitors

KIT is recognized as one of the rational therapeutic targets. The KIT gene is an oncogene that encodes a transmembrane glycoprotein belonging to the tyrosine kinase (PTK) receptor family [79]. KIT mutations have been identified in more than 35% of acral and mucosal melanomas [80]. The KIT mutation lead to spontaneous ligand-independent receptor dimerization of KIT protein, stimulating excessive cell proliferation and anti-apoptotic signaling, thus playing an important role in MM growth [81,82]. In a genetic analysis of MM patients conducted in 2011, it was shown that 17% of MM patients in China develop the disease due to C-Kit mutation. Therefore, targeted therapies for C-Kit mutation play a key role in the treatment of Chinese MM patients [83]. The National Comprehensive Cancer Network (NCCN) treatment guidelines approved Imatinib as a guideline drug for the treatment of MM caused by KIT mutation in 2013 through a study of its therapeutic effects and side effects [84].

A phase II clinical trial conducted in China reported for the first time that Imatinib was effective in prolonging the PFS of advanced metastatic MM patients caused by KIT mutations and in improving the rate of disease control and treatment efficiency [51]. In addition, in some phase II clinical trials of mucosal MM caused by KIT mutations, the CR in the Dasatinib-treatment group, which is also a Kit inhibitor, reached only 18%, a significant decrease compared to Imatinib-treatment [52]. Therefore, even though KIT inhibitors are subject to more side effects in practice, Imatinib is currently still the first choice for targeted treatment of metastatic mucosal MM [66].

### 3.4. Indoleamine 2,3-Dioxygenase (IDO1) Inhibitors

IDO1 is the catalytic rate-limiting enzyme of the first step of the major metabolic pathway of L-tryptophan (L-Trp) [85], and IDO1 overexpression can lead to L-Trp depletion in the local microenvironment and subsequent impaired T-cell function [86,87]. In recent years, several new structural types of IDO1 inhibitors have been discovered, and a few have entered the clinical stage, such as Indoximod [88], Epacadostat [89] and Navoximod [90], but no successful drugs have been marketed yet.

Tryptophan-derived inhibitors were among the first reported IDO1 inhibitors [91], the most representative of which is Indoximod. The results of a clinical trial showed that the combination of Indoximod and the anti-PD-1 antibody Pembrolizumab in advanced MM patients had a PR of 61%, a median PFS of 12.9 months, and a 1-year survival rate of 56%, suggesting that Indoximod and Pembrolizumab have a significant synergistic anti-tumor effect [92].

Epacadostat is a competitive substrate for the L-Trp, which is an IDO1 substrate [93]. Epacadostat can interfere with the abnormal metabolism of Trp in tumor cells by competing for binding to the enzymatically active catalytic structural domain of IDO1 [94], thereby inhibiting tumor growth. Data from clinical trials in MM showed that Epacadostat in combination with the anti-CTLA-4 antibody Ipilimumab not only reduced tumor size but also improved disease control rates and PFS in patients [95]. Meanwhile, the combination of the anti-PD-1 antibody Pembrolizumab and Epacadostat was tried in a randomized phase III study. Despite having better clinical outcomes in early trials, Epacadostat failed to prevent IDO1-dependent MM immune escape and did not increase PFS and OS when treated with Pembrolizumab monotherapy [96].

Navoximod is a small molecule inhibitor of IDO1 under investigation [97]. In vivo studies have shown that oral administration of Navoximod reduces Kynurenine concentrations in plasma and tissues by approximately 50% and causes effector T cells to show dose-dependent activation and proliferation, thereby reducing the size of MM [98]. Phase I clinical trials of Navoximod alone and in combination with the anti-PD-L1 antibody Atezolizumab for the treatment of solid tumors have been completed, but no published data are available at this time [99].

BMS-986205 is a novel IDO1 inhibitor developed by Bristol-Myers Squibb [100]. Preclinical trials have demonstrated dose-dependent efficacy of BMS-986205 [101]. Though BMS-986205 successfully inhibited IDO1 and reduced human canine urinary quinolinic acid serum levels even at low concentrations, it had better efficacy and pharmacokinetics compared to Epacadostat [102].

### 3.5. VEGFR (Vascular Endothelial Growth Factor Receptor) Inhibitors

VEGF is involved in regulating processes such as angiogenesis development and embryopoiesis, and it has been found that the production of VEGFR-1 and laminin is required for tumor growth. VEGFR and its family members are often at abnormal levels in malignant tumors [103,104]. A growing number of preclinical and clinical studies have demonstrated the role of VEGF signaling in melanoma progression, treatment response, and OS [105,106]. The results of clinical trials showed that the novel VEGFR inhibitor Vatalanib was able to delay disease progression in a subset of patients, although it did not significantly improve OS of patients [107]. Another clinical trial showed that Axitinib, a second-generation inhibitor of VEGFR, was effective in inhibiting the progression of stage II metastatic MM and was well tolerated by patients [108]. Another trial demonstrated that Axitinib in combination with Paclitaxel and Carboplatin was effective in prolonging OS in MM patients [109].

## 4. Combination Therapy

Combination therapy has substantially improved clinical outcomes in patients with metastatic MM, with approximately 50% of patients responding to treatment [110]. The use of immunotherapy and targeted therapy has broken the limitations of conventional MM treatment, but both immunotherapy and targeted therapy have shown resistance and toxicity in the treatment of MM. In order to better address this problem, trials on their combination regimens have been conducted at this stage (Figure 6).

The combination of Ipilimumab and Nivolumab is widely recognized as the most effective first-line treatment for patients with advanced MM [101]. The combination of the MEK inhibitor Cobimetinib and BRAF inhibitor Vemurafenib with anti-PD-1 antibody Atezolizumab showed better PFS in patients with metastatic MM than those treated with Cobimetinib and Vemurafenib [111]. The safety shown by the three-drug combination group during treatment is consistent with the known safety of each drug. As a result, Cobimetinib + Vemurafenib + Atezolizumab became the first three-drug combination group approved by the FDA for patients with BRAF mutation, unresectable or metastatic MM on 30 July 2020. In addition, clinical trial results for the three-drug combination of Dabrafenib + Trametinib + Pembrolizumab showed that the three-drug combination group improved PR and CR in MM patients with resectable stage III BRAF-mutant [112].

Only one three-drug combination group has been approved by the FDA, but several three-drug combination group regimens have entered clinical trials. For example, the stage II clinical trial for the anti-PD-1 antibody Camrelizumab in combination with the VEGFR inhibitor Apatinib and the chemotherapeutic agent Temozolomide in patients with advanced limb MM noted that the 66.7% objective response rate demonstrated by this combination regimen over the course of treatment exceeds that of any other drug combination explored in the literature. With a significant improvement in PFS, the regimen is expected to meet the urgent international need for treatment options for limb-end MM [113].

Trials for the tryptophan inhibitor ADI PEG 20 + PD-1 monoclonal antibody Nivolumab + CTLA-4 monoclonal antibody Ipilimumab in patients at high risk of metastatic uveal MM and for the PD-1 monoclonal antibody Spartalizumab + BRAF inhibitors Dabrafenib and Trametinib in patients with unresectable or metastatic BRAF V600 mutation-positive cutaneous MM have also concluded [114,115]. Although the study did not meet the primary endpoint of PFS, it did provide additional reference data for the development of combination regimens for the treatment of MM. The results of these trials suggest that three-drug combination therapy may be the newest approach for the treatment of metastatic MM, but longer follow-up is needed because the results are still immature [111].

## 5. Conclusions

MM is characterized by an insidious onset, aggressiveness, and poor prognosis. In recent years, immunotherapy and targeted therapy have emerged as important treatment modalities. They can indeed improve the prognosis of patients with advanced MM to some extent. In patients with primary MM, these therapies also contribute to improved survival. However, when it comes to some advanced patients with metastases, the situation is far from satisfactory. One of the main problems is that patients may suffer from varying degrees of drug resistance. In addition, irAEs are an issue. Combination therapy, which combines different treatments, such as immunotherapy and targeted therapy or other traditional therapies, has made great advances. Combination therapy has the potential to overcome some of the limitations of single-agent therapy. The effectiveness of the treatment, in terms of whether it can truly and consistently deliver better results in a larger patient population, still needs further research. Similarly, the safety of combination therapy needs to be more fully assessed through clinical validation. Future studies will focus on new immunotherapy targets and the combined use of multiple treatments, which may lead to more personalized and effective MM treatment, better clinical trial options, and improved overall survival and quality of life.

## Figures and Tables

**Figure 1 biomedicines-13-00225-f001:**
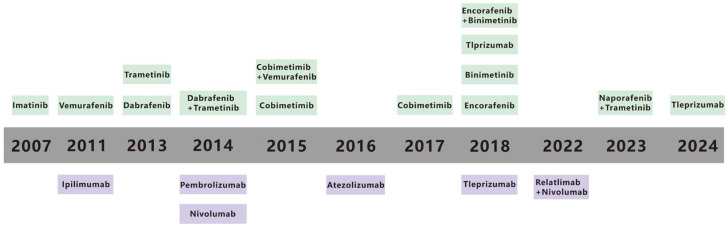
Timeline for FDA-approved therapies for metastatic melanoma. The timeline highlights key advancements in the fields of immunotherapy and targeted therapy from 1990 to 2021. Therapies are categorized into immunotherapies (in purple) and targeted therapies (in green), reflecting the evolution of treatment strategies over time. Reproduced from ref [15].

**Figure 2 biomedicines-13-00225-f002:**
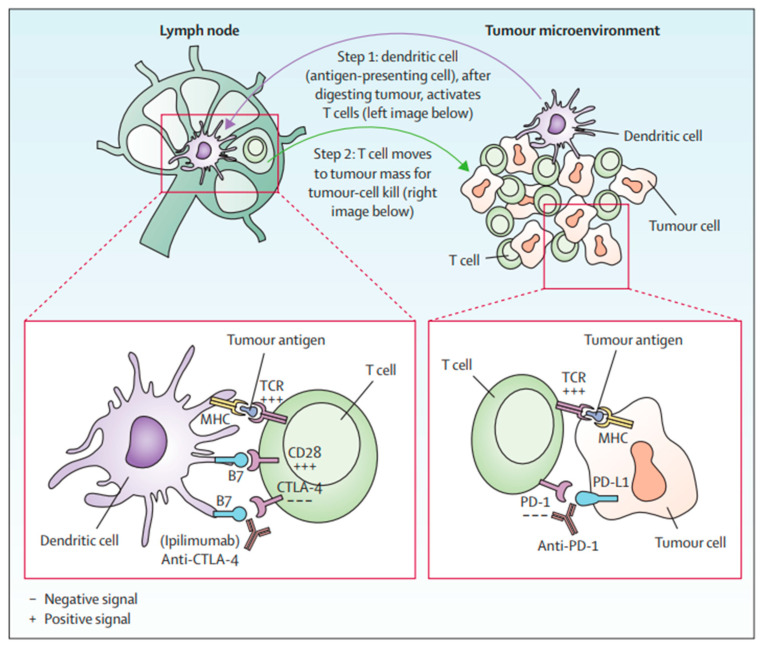
T-cell activation by anti-CTLA-4 and anti-PD1 [12].

**Figure 3 biomedicines-13-00225-f003:**
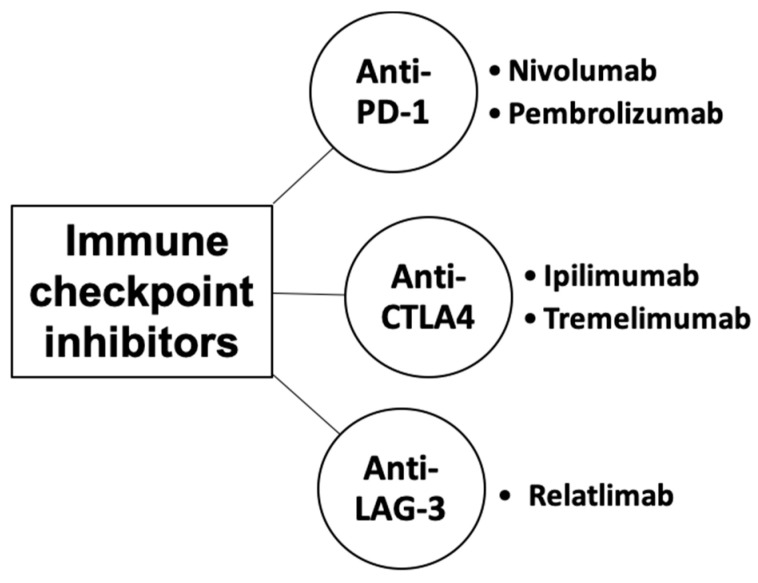
Classes of immune checkpoint inhibitors for MM.

**Figure 4 biomedicines-13-00225-f004:**
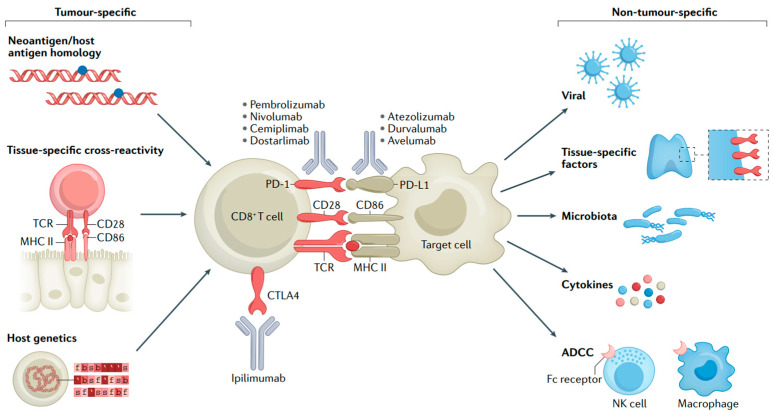
Mechanisms of irAEs. T-cell interactions with malignant or non-malignant cells and molecular mechanisms of immune checkpoint blockade [56].

**Figure 5 biomedicines-13-00225-f005:**
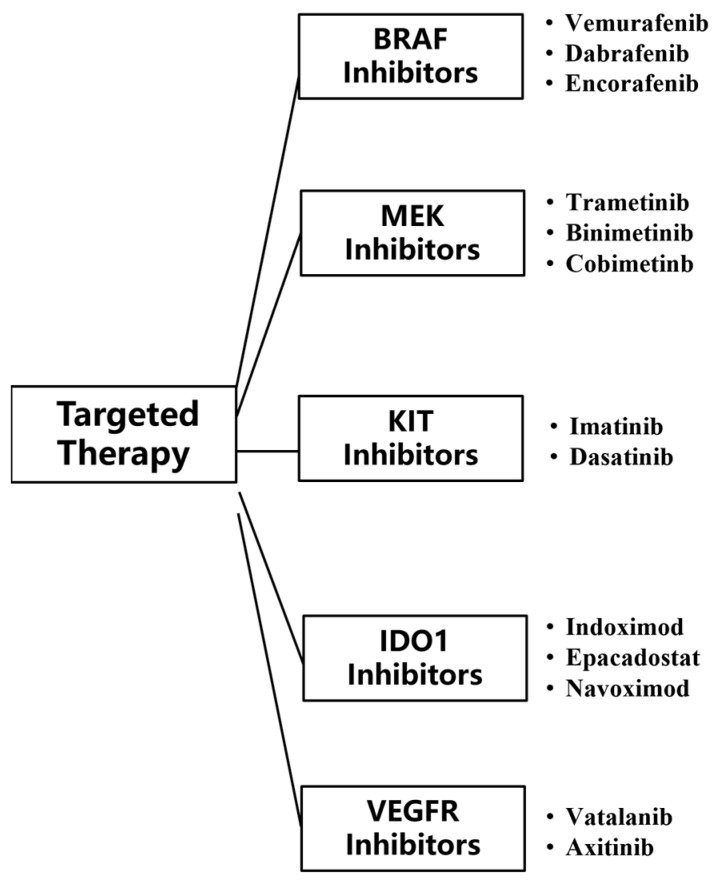
Classes of targeted therapy checkpoint inhibitors for MM.

**Figure 6 biomedicines-13-00225-f006:**
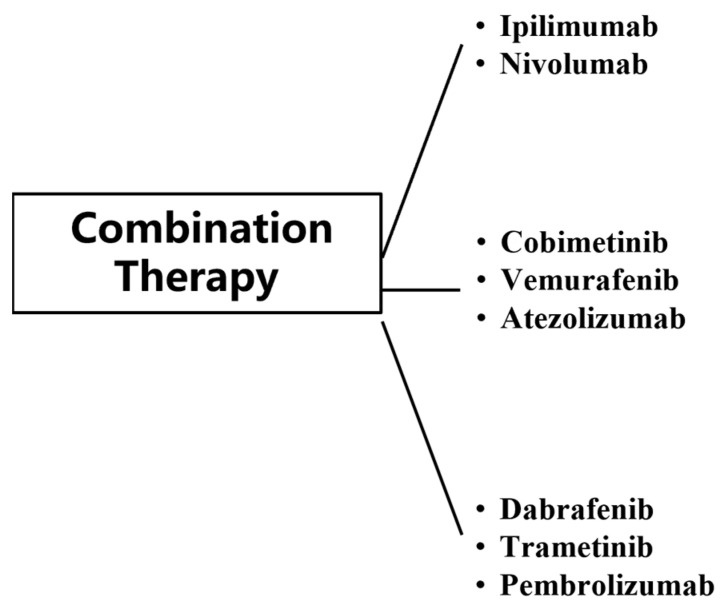
Classes of combination therapy for MM.

## Data Availability

Not applicable.

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
