# Peer review of "Advances in Immunotherapy and Targeted Therapy of Malignant Melanoma"

_biomedicines, 2025, doi:10.3390/biomedicines13010225_

Round 1

Reviewer 1 Report

Comments and Suggestions for Authors

In this review the authors give a quite complete overview on the more recent applications of immunotherapy and targeted therapy in advanced melanoma.

The review is well organized, correctly focused and based on a rich literature.

Minor points:

The review is generally correctly written. Only the “Conclusion” section needs some little revisions.

Line 297:   Nevertheless, immunotherapy …

Line 299  ...for some patients with advanced metastasis…

Line 304  …and on the combined …

Author Response

Comments1: Comments and Suggestions for Authors

In this review the authors give a quite complete overview on the more recent applications of immunotherapy and targeted therapy in advanced melanoma.

The review is well organized, correctly focused and based on a rich literature.

Minor points:

The review is generally correctly written. Only the “Conclusion” section needs some little revisions.

Line 297:   Nevertheless, immunotherapy …

Line 299  ...for some patients with advanced metastasis…

Line 304  …and on the combined …

Response: Thanks for your insightful comments and suggestions regarding our manuscript. We have revised the “Conclusion” section based on your comments, and the part we have modified were marked in the manuscript.

In “Conclusion” section:

MM is characterized by an insidious onset, aggressiveness and poor prognosis. In recent years, immunotherapy and targeted therapy have emerged as important treatment modalities. They can indeed improve the prognosis of patients with advanced MM to some extent. In patients with primary MM, these therapies also contribute to improved survival. However, when it comes to some advanced patients with metastases, the situation is far from satisfactory. One of the main problems is that patients may suffer from varying degrees of drug resistance. In addition, irAEs are an issue. Combination therapy, which combines different treatments such as immunotherapy and targeted therapy or other traditional therapies, has made great advances. Combination therapy has the potential to overcome some of the limitations of single-agent therapy. The effectiveness of the treatment, in terms of whether it can truly and consistently deliver better results in a larger patient population, still needs further research. Similarly, the safety of combination therapy needs to be more fully assessed through clinical validation. Future studies will focus on new immunotherapy targets and the combined use of multiple treatments, which may lead to more personalised and effective MM treatment, better clinical trial options, and improved overall survival and quality of life.

Reviewer 2 Report

Comments and Suggestions for Authors

In their manuscript "Advances in immunotherapy and targeted therapy of malignant melanoma" the author review a very intersetung topic in medicine which is still an unmet medical need.

The current state of research in this field is well stated.

However, as this is a review and mentiones therapies, which are not easily to understand to all of the readership and only to experts (but the journal "biomedicines" has a broad readership - and an high impact factor . I recommend urgently to revise the manuscript to improve the quality of the review:

- 1. Introduction: As the authors state correctly, thre malignant melanoma accounts a high mortality.
The authors should als state, which organs commonly affected by the metastasis (which causes finally the mortality). Thus , therapie should als be effective to the metastases

- 2. The immunotherapy and Inhibitors
Please explain how Immune checkpoint inhibitors act. Porvide an understandable sketch which shows the mechanism (so explain  e.g how NKC´s "recognize" the cell to be attacked.
what are disadvantages in the use of Checkpoint inhibitors?

-3- Targeted Therapy .....
Also probode a graphical sketch to explain
The manuscript needs some workup to be a good review (and not only a "written" summary.

-3.5 : same

The work is worth publishing after revison and taking the above mentioned points to be a good review. Its need sketches which visualize the written words. So, at least 4-5 figures should be added.

Author Response

Comments 1: Introduction: As the authors state correctly, thre malignant melanoma accounts a high mortality.
The authors should als state, which organs commonly affected by the metastasis (which causes finally the mortality). Thus , therapie should als be effective to the metastases

Response 1: We sincerely thank the reviewer for the valuable comment. The reviewer is correct in pointing out the importance of stating the commonly affected organs by metastasis in MM, as this is crucial in understanding the high mortality rate associated with this disease. We have added a section in the Introduction to discuss the organs that are commonly affected by the metastasis of MM. These typically include the lungs, liver, brain, lymph node, and bones. The spread of melanoma cells to these organs often leads to the development of multiple organ failure and ultimately contributes to the high mortality rate. Regarding the mention of therapy effectiveness against metastases, we will also touch upon this briefly in the Introduction, stating that current research is focused on developing therapies that can not only target the primary tumor but also effectively treat metastases, as this is a key aspect in improving the survival rate of patients with malignant melanoma. We have made changes in the manuscript.

Comments 2:  The immunotherapy and Inhibitors
Please explain how Immune checkpoint inhibitors act. Porvide an understandable sketch which shows the mechanism (so explain  e.g how NKC´s "recognize" the cell to be attacked.
what are disadvantages in the use of Checkpoint inhibitors?

Response 2: We sincerely thank the reviewer for the valuable comment. Immune checkpoint inhibitors (ICIs) are a type of immunotherapy that work by blocking proteins on immune cells or tumor cells that prevent the immune system from attacking cancer cells. These proteins, known as immune checkpoints, are normally involved in regulating the immune response to prevent it from attacking healthy cells. However, cancer cells can use these checkpoints to evade the immune system. One of the most well-known immune checkpoints is the programmed death-1 (PD-1) receptor on T cells, which interacts with its ligand, PD-L1, on tumor cells. When PD-1 and PD-L1 bind together, it sends a signal to the T cell to stop attacking, effectively shutting down the immune response. ICIs, such as anti-PD-1 or anti-PD-L1 antibodies, block this interaction, allowing the T cells to remain active and continue to attack the tumor cells. Another important checkpoint is cytotoxic T-lymphocyte-associated protein 4 (CTLA-4), which also regulates T cell activation. CTLA-4 inhibitors work by preventing CTLA-4 from binding to its ligands, CD80 and CD86, on antigen-presenting cells, thereby enhancing T cell activation.

Despite Immunotherapy have achieved remarkable success in MM treatment, they also have certain disadvantages. Firstly, ICIs may lead the immune system to attack other organs in the body, resulting in immune-related adverse events (irAEs) like pneumonia, hepatitis, and endocrine diseases. Secondly, there is the issue of drug resistance. Some patients may become resistant to ICIs, which causes a decline in treatment effectiveness. Furthermore, the high cost of these inhibitors may limit their use among some patients. Future studies need to further explore ways to enhance the efficacy of these drugs, reduce adverse effects, and cut down the treatment cost.

We added a figure to explain how Immune checkpoint inhibitors act and the  disadvantages in the use of Checkpoint inhibitors in the manuscript.

Comments 3:  - Targeted Therapy .....
Also probode a graphical sketch to explain
The manuscript needs some workup to be a good review (and not only a "written" summary.

-3.5 : same

The work is worth publishing after revison and taking the above mentioned points to be a good review. Its need sketches which visualize the written words. So, at least 4-5 figures should be added.

Response 3: We sincerely thank the reviewer for the valuable comment. According to the reviewer's kindly suggestion, we added 4 figures to visualize the written wordsin the manuscript.

Reviewer 3 Report

Comments and Suggestions for Authors

The manuscript entitled “Advances in immunotherapy and targeted therapy of malignant melanoma by Xue Wang et al. provides a comprehensive overview of the current development status and research progress of immune checkpoints, targeted genes, and their inhibitorÑ–, with a view to providing a reference for the clinical treatment of malignant melanoma. The manuscript may be of general interest to the researchers of this field, but the manuscript lacks some information that the author should consider and incorporate in the present form of the manuscript. Here are concerns that need to be addressed in the present form of the manuscript.

 1.     The introduction is too short and should be extended. More references to recent publications should be added. The inclusion of these publications could strengthen the novelty and relevance of the work.

2.     A comparative analysis of different therapies using meta-analytical methods could provide a more robust evaluation of their relative efficacy and safety.

3.     The conclusions section is overly general.

4.  Minor typographical errors, such as "FAD" instead of "FDA," need correction.

Author Response

Comments 1: The introduction is too short and should be extended. More references to recent publications should be added. The inclusion of these publications could strengthen the novelty and relevance of the work.

Response 1: We really appreciate the Reviewer's valuable comments. We have extended the introduction and added some recent publications, in order to strengthen the novelty and relevance of the work.

Comments 2: A comparative analysis of different therapies using meta-analytical methods could provide a more robust evaluation of their relative efficacy and safety.

Response 2: We really appreciate the Reviewer's valuable comments.A comparative analysis of different therapies using meta-analytic methods can indeed provide a more robust assessment of their relative efficacy and safety. However, given that meta-analysis requires a large amount of data from multiple trials, and given the differences in patient samples in different trials, it becomes difficult to conduct a meta-analysis. therefore, we have not analysed this part for the time being, and will further organise and analyse the data in the future.

Comments 3: The conclusions section is overly general.

Response 3: Thanks for the Reviewer's valuable comments. We have revised the conclusion section.

Comments 4:  Minor typographical errors, such as "FAD" instead of "FDA," need correction.

Response 4: We sincerely apologize for our oversight, and we have checked and revised the full manuscript.